# Pathological Changes in Pancreatic Carcinogenesis: A Review

**DOI:** 10.3390/cancers13040686

**Published:** 2021-02-08

**Authors:** Keiko Yamakawa, Juanjuan Ye, Yuko Nakano-Narusawa, Yoko Matsuda

**Affiliations:** Oncology Pathology, Department of Pathology and Host-Defense, Faculty of Medicine, Kagawa University, 1750-1 Ikenobe, Miki-cho, Kita-gun, Kagawa 761-0793, Japan; yamakawa@med.kagawa-u.ac.jp (K.Y.); jj528575963@gmail.com (J.Y.); y_nakano@med.kagawa-u.ac.jp (Y.N.-N.)

**Keywords:** pancreatic cancer, pathology, carcinogenesis

## Abstract

**Simple Summary:**

Pancreatic cancer has an extremely poor prognosis. Pathological characteristics of pancreatic carcinogenesis, including precancerous lesions and cancers, might provide valuable information for the development of early diagnosis and effective treatments. Autopsy studies have revealed pathological characteristics of precancerous lesions. Animal studies using hamsters and mice have revealed the mechanisms of carcinogenesis. We have summarized pathological changes in the pancreas of humans and experimental animals.

**Abstract:**

Despite advances in diagnostics and therapeutics, the prognosis of pancreatic cancer remains dismal. Because of a lack of early diagnostic methods, aggressive local progression, and high incidence of distant metastasis, most pancreatic cancers are inoperable; therefore, the characteristics of early pancreatic cancer have not been well understood. Autopsy studies revealed the characteristics of prediagnostic pancreatic malignancies, including precancerous lesions, early stage pancreatic cancer, and pancreatic cancer without clinical symptoms (occult cancers). Animal models using hamsters and genetically engineered mice have focused on mechanisms of carcinogenesis, thereby providing insights into risk factors and prevention and serving as a preclinical test for the development of novel diagnostic and treatment modalities. In this review, we have summarized pathological changes in the pancreas of humans and experimental animals during carcinogenesis.

## 1. Introduction

The annual incidence of pancreatic ductal adenocarcinoma (PDAC) has been increasing, and it is the leading cause of cancer-related deaths worldwide [1,2]. Tobacco use, heavy alcohol consumption, diabetes, obesity, pancreatitis, vitamin D deficiency, solar radiation, aging, and family history are major risk factors for the incidence of PDAC [3]. Despite advances in diagnostics and therapeutics, the prognosis of PDAC remains poor, with an overall 5-year survival rate of below 10% [4,5]. Because of the lack of early diagnostic methods, aggressive local progression, and high incidence of distant metastasis, approximately 70% of PDACs are diagnosed at advanced stages when patients present with symptoms. Most PDACs are inoperable because the characteristics of early stage PDACs are not well understood. Prediagnostic pancreatic malignancies, including precancerous lesions, early stage PDAC, and PDAC without clinical symptoms (occult cancers), can be detected only on autopsy [6].

Recent molecular studies have shown that the development of PDAC occurs at least in part through the intraepithelial proliferation/dysplasia–cancer sequence. Approximately 90% of PDACs can arise from pancreatic intraepithelial neoplasia (PanIN), with a small proportion arising from intraductal papillary mucinous neoplasm (IPMN) or mucinous cystic neoplasm (MCN). Recently, it has been reported that the centro–acinar region might be the origin of PDAC, and PanINs or atypical epithelium is considered to arise in the acinar–ductular transformation [7,8]. Among them, high-grade PanINs might be an immediate precursor to PDAC. The potentially curative treatment is radical surgical resection. However, improvements in overall survival after resection have been observed only in cases in which the tumor is small and confined to the pancreas or, ideally, pre-invasive [7,8]. Thus, early diagnosis of PDAC, especially in its pre-invasive stage, that is, high-grade PanIN (carcinoma in situ), is most important [9].

Many studies that have used genetically engineered mice have been published since the 2000s, and studies have used the Syrian hamster model since its development in the mid-1970s. These studies focus on the mechanisms of carcinogenesis, thereby providing insights into risk factors and prevention and serving as a preclinical test for the development of novel diagnostic and treatment modalities [10]. In this review, we have summarized pathological changes in the pancreas of humans and experimental animals during carcinogenesis.

## 2. Pathology of Pancreatic Tumors

### 2.1. Primary Pancreatic Tumors

Autopsy studies have shown that the incidence of macroscopic pancreatic lesions is 6% [11]. Primary pancreatic tumors are observed in 3% of cases, and most primary tumors are PDACs (2%), followed by neuroendocrine tumors (0.3%) and intraductal papillary mucinous neoplasms (0.1%). High-grade PanINs are detected in 0.04% of the cases. Occult PDAC is discovered incidentally in 8% of PDAC cases [11].

PDAC is characterized by a robust fibro-inflammatory response, namely, desmoplastic reaction [12]. The desmoplastic reaction is the result of a complex interplay of cancer cells, vascular cells, pancreatic stellate cells, and inflammatory cells, and it promotes proliferation, invasion, and chemoresistance in PDAC cells [13]. Distant metastasis is an important prognostic factor for PDAC. Twenty-seven percent of occult cancer cases involve distant metastasis, and most of them are localized in the pancreatic tail. Asymptomatic cancers of the pancreatic tail have a greater ability to metastasize to distant sites than those of the pancreatic head and body. These results indicate the aggressive features of PDACs.

### 2.2. Secondary Pancreatic Tumors

Secondary pancreatic tumors resulting from the invasion and metastasis of malignant tumors were detected in 175 cases (2%). The primary sites of metastatic tumors were the stomach (17%), lung (17%), colorectum (6%), esophagus (5%), gall bladder (3%), and kidney (2%). The primary sites of direct invasion were the stomach (19%), common bile duct (3%), gall bladder (1%), and liver (1%) [11]. The infiltration rate of hematologic tumors was 16%. This indicates that we should pay attention to various primary and secondary tumors in the pancreas.

## 3. Pathology of Precancerous Lesions of the Pancreas

### 3.1. Cancer-Related Lesions of the Pancreas

Cancer-related lesions are classified into precancerous lesions and surrogate markers of cancers (Table 1) [14]. ADM, cyst, PanIN, and IPMN (Figure 1A–D) are precancerous lesions of PDAC, whereas fatty replacement and lobulocentric atrophy (Figure 1E,F) are surrogate markers of PDAC. It is difficult to detect PanIN lesions using currently available imaging modalities, including endoscopic ultrasound (EUS), computed tomography, and magnetic resonance imaging; therefore, we need to establish imaging indicators of higher grade PanIN for the identification of PDAC at an early stage (Figure 1G,H).

### 3.2. Pancreatic Intraepithelial Neoplasia (PanIN)

Previous autopsy studies have reported that the incidence of high-grade dysplasia/carcinoma in situ in the pancreas was 2.9–4.0% [15,16,17,18]. In contrast, high-grade PanIN in patients with PDAC was 40% [19]. We evaluated PanIN lesions in 173 consecutive autopsy cases without PDAC or IPMN by subjecting the entire pancreas for microscopic examination [15]. High-grade PanIN was found in 4% of examined cases associated with a high incidence of diabetes mellitus and/or older age. High-grade PanINs were always multifocal, and the number of high-grade PanIN foci was positively associated with that of low-grade PanIN. High-grade PanIN was located more frequently in the pancreatic body and tail than in the pancreatic head and predominantly involved small interlobular/intralobular ducts rather than the main duct.

Microscopically, aged pancreas shows a decrease in pancreatic parenchymal cells, including endocrine (islet) and exocrine (acinar and duct) cells, resulting in the induction of oxidative stress. They are replaced by fat cells and fibroblasts. The decreased number of pancreatic parenchymal cells is closely associated with pancreatic dysfunction. A lot of studies indicate that PanINs are usually accompanied by fibrosis [18]. Both dysplasia and hyperplasia of the pancreatic ducts are often associated with fibrosis, especially in the pancreatic body and tail [20]. Multicentric PanINs are associated with lobular atrophy of the pancreatic parenchyma and chronic pancreatitis [21]. A study involving a familial PDAC cohort indicated that pancreatic parenchymal atrophy might be secondary to the obstruction of small pancreatic ducts by the proliferative epithelium of PanIN [21]. Fibrosis plays a key role in carcinogenesis steps of the pancreas, via the production of cytokines and extracellular matrices by fibroblasts [22]. Furthermore, an increasing PanIN grade was associated with fibrosis and chronic pancreatitis on EUS [9]. PanINs were associated with fat infiltration of the pancreas, intralobular fibrosis, a high BMI, and subcutaneous and intravisceral fat [23].

### 3.3. Pancreatic Cystic Lesions

Pancreatic cyst is one of the precursors of PDAC. IPMNs are the most common type of pancreatic cystic neoplasms, accounting for at least 20% of all resected pancreatic cystic neoplasms. Other than IPMN, there are various cystic lesions, such as MCN, serous cystic neoplasm, duct ectasia, and macrocysts with a mucinous lining [14,24].

We analyzed the pancreas in 280 consecutive autopsy cases without PDAC or IPMN by subjecting the entire pancreas for microscopic examination [15]. We found 236 cysts in 93 patients (33%), and 9 cysts (4%) had high-grade dysplasia, similar to the result of a previous report (3%) [16]. In contrast, there were 15 non-cystic lesions with high-grade dysplasia. In total, 24 high-grade dysplastic lesions were noted in 15 patients (5%). Of the 15 patients with high-grade dysplastic lesions, in 10 patients, the condition was accompanied by pancreatic cysts. Patients with cysts showed a higher incidence of high-grade dysplasia (11%; *p* = 0.005) than patients without cysts (3%). All cysts with high-grade dysplasia were located in the branch duct of the pancreatic head/body, while 20% of non-cystic lesions with high-grade dysplasia were located in the main pancreatic duct. This suggests that cystic lesions with high-grade dysplasia may have characteristics different from those of non-cystic high-grade dysplasia.

The incidence of cystic and non-cystic high-grade dysplasia is low; therefore, cysts themselves are less likely to be a precursor of PDAC. However, rare cysts have malignant potential. Patients with multiple and large cysts may have an increased risk of developing into PDAC. Furthermore, cystic high-grade dysplasia might include incipient IPMNs of the gastric type with high-grade dysplasia.

Various carcinogens, environmental factors, and aging-related telomere dysfunction contribute to accumulation of genetic abnormalities [25]. These genetic events, oxidative stress, and parenchymal atrophy may accelerate carcinogenesis, and obstruction of the pancreatic ducts due to proliferative epithelial lesions and/or inflammation may induce retention cyst formation. Secretion of abundant mucin by hyperplastic or metaplastic duct epithelium may also induce pancreatic cyst formation.

In a recent study via radiological imaging and pancreatic juice cytology, a high-grade PanIN presented with a localized stricture of the main pancreatic duct [26]. However, high-grade PanINs that underwent autopsy are often detected in the branch pancreatic duct [15]. There might be differences in pathogenesis and risk factors between high-grade dysplasia involving branch ducts, resulting in cystic changes vs. high-grade dysplasia involving the main pancreatic duct, leading to stenosis of the main duct. Small branch duct lesions are usually asymptomatic. It is important to develop new laboratory tests to detect these lesions.

## 4. Animal Models of Pancreatic Carcinogenesis

### 4.1. Hamster

Animal models are necessary to develop early diagnostic tools and explore new therapeutic approaches. Syrian golden hamster has been used as an animal PDAC model, as it reflects the human physiology. Multiple chemicals have been used to model tumorigenesis [27] (Table 2). Subcutaneous injection of N-nitrosobis(2-oxopropyl)amine (BOP) had induced PDACs in up to 90% of animals within 3–12 months, the histology of which is similar to that of human tumors [28]. An initial subcutaneous injection of BOP 70 mg/kg body weight, followed by three cycles of ethionine-methionine rescue-induced pancreatic regeneration 12 days later produced pancreatic carcinoma after 10 weeks [29]. Ethanol and NNK induced PanIN and PDAC at over 50% incidence [30]. Furthermore, bile reflux into the pancreatic ducts produced intraductal papillary carcinoma in hamsters [31]. K-ras activation by point mutations and p16 inactivation by aberrant methylation of 5′ CpG islands or by homologous deletions have been observed in both hamster and human PDACs [32]. The hamster model has been used to validate and dissect numerous conditions thought to modulate human cancer risk [32].

### 4.2. Mouse

The first genetically engineered PDAC mouse model was *Pdx1*-Cre; LSL-*Kras*^G12D^ mice [27,33] (Table 2). It expressed an oncogenic *Kras*^G12D^ allele from its endogenous promoter in the developing pancreas through Cre-mediated recombination driven by *Pdx1* regulatory elements. Pdx1 is a homeodomain protein critical for early pancreatic development, and both differentiated exocrine and endocrine cell types in the mature pancreas arise from a Pdx1-expressing progenitor population. It developed PanIN lesions, and a minor proportion of them (<10%) developed invasive metastatic adenocarcinomas after several months.

Complex *Kras-*based genetically engineered mouse models showed a higher frequency of metastasis than basic *Kras-*based genetically engineered mouse models [27]. Engineering mice that express oncogenic *Kras* in the pancreas in combination with a conditional Ink4a/Arf of the *p53*-null allele [34,35] or a conditional knock-in mutant *p53*^R172H^ [36] developed lethal metastatic PDACs with complete penetrance and shorter latency. Engineering mice that express oncogenic *Kras* in the pancreas in combination with a conditional *SMAD4* [37,38], *Notch1* [39], or *Tgfbr2*-null allele [40] or a conditional knock-in mutant *Brca2* [41] also developed metastatic PDACs.

Engineering mouse models provide insights into the origin of PDAC. Activating *Kras*^LSL^ in adult acinar cells produces PanIN lesions [42]. Acinar-derived PanINs progressed to invasive PDAC in the context of *Tp53* depletion [43]. In contrast, activating *Kras*^LSL^ in adult duct cells produced almost no PanINs [44]. These reports indicate that PDAC originates from differentiated acinar cells and non-ductal cells. Furthermore, acinar-derived PanINs in the *Kras*^LSL^ model do not show acinar characteristics, suggesting that lesion formation requires extensive phenotype reprogramming, named acinar–ductal reprogramming [10,45]. Inflammation might enhance ductal reprogramming of *Kras* mutant acinar cells. Acinar–ductal reprogramming occurs in mutant acinar cells, and it is different from acinar–ductal metaplasia, which is the transient upregulation of duct markers in wild-type acinar cells in response to injury. However, the presence or absence of acinar–ductal reprogramming in the human pancreas is unclear, warranting further studies.

### 4.3. Other Models

Genetically engineered animal models accurately mimic the pathophysiological features of human PDAC including initiation from precancerous lesions and containing cancer cells, stromal components, and immune cells. However, it usually requires a long tumor latency period. It is often difficult to obtain appropriate genetically engineered animal models. Therefore, several alternative methods have been developed.

Xenograft is the transplantation of living cells, tissues, or organs from one species to another. Usually, human-derived cancer cells or cancer tissue are implanted into immunodeficient rodents. Xenograft models using pancreatic satellite cells can show desmoplasia, which is the main characteristic feature of PDAC [46]. It can provide short tumor latency and uniformity in the initial tumor burden, but it lacks a functional immune system [46]. Furthermore, cancer cells and stromal components belong to different species. More recently, the development of patient-derived xenografts has provided great benefit for the development of personalized therapies since drugs could be directly tested on each individual patient’s tumor. The PDX model has facilitated translational studies on PDAC.

Allograft is the transplantation of living cells, tissues, or organs between the same species. It can provide cancer tissues with heterogeneous stromal components and a functional immune system, but all the components are of non-human origin [46].

There are various transplanted tumor models, such as orthotopic or ectopic implantation; metastasis model of the liver, lung, or lymph node; and perineural invasion model [40]. However, the establishment of transplanted tumor models usually requires effort and time as well as materials. Therefore, organoid models of PDAC have been developed to mimic human PDAC [47,48]. It can control and reproduce the three-dimensional structure of PDAC in the short term. It can be used to study tumorigenesis, tumor development, and treatment effects for each patient and provide great benefit to develop individualized treatments.

## 5. Conclusions

Various experimental models have been used to study the features of PDAC. Pathological changes in the carcinogenesis of the pancreas are important, and they aid in understanding the mechanisms of PDAC progression and developing novel strategies for the early detection and treatment of PDAC. We need to develop a clinical test for early PDAC to improve the prognosis of PDAC.

## Figures and Tables

**Figure 1 cancers-13-00686-f001:**
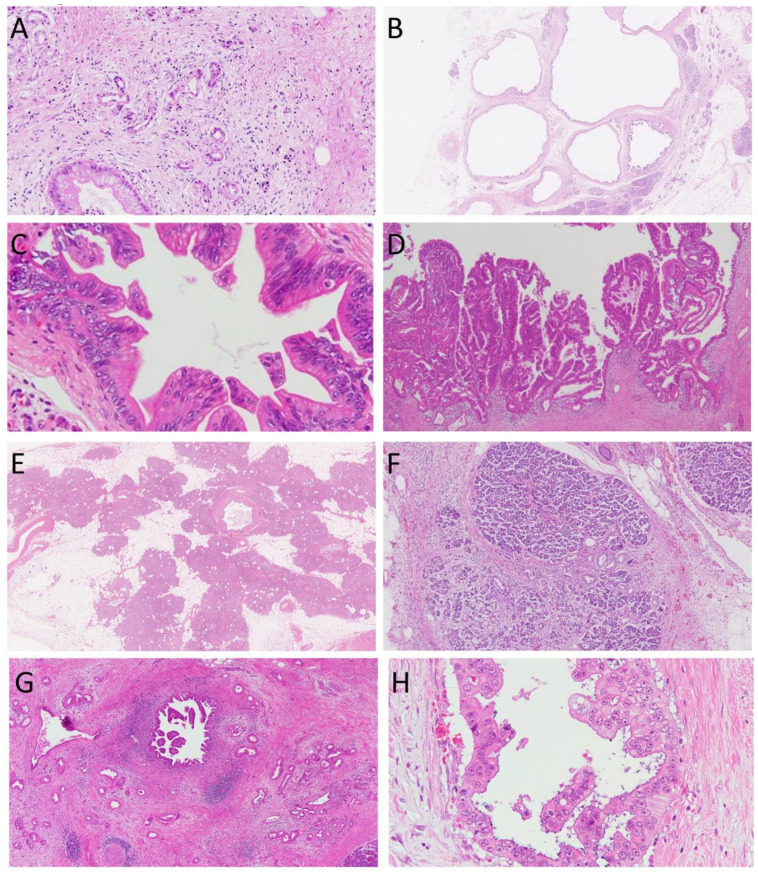
Cancer-related lesions of the pancreas. (**A**), acinar-to-ductal metaplasia, (**B**), cyst, (**C**), pancreatic intraepithelial neoplasia, (**D**), intraductal papillary mucinous neoplasm, (**E**), fatty replacement, (**F**), lobulocentric atrophy, (**G**), and (**H**), pancreatic ductal carcinoma. Hematoxylin and eosin staining. Original magnification, (**A**,**C**,**H**), ×400; (**B**,**E**), ×40; (**D**,**F**,**G**), ×100.

**Table 1 cancers-13-00686-t001:** Cancer-related pathological changes in the human pancreas.

Precursor lesions of pancreatic cancer
Acinar-to-ductal metaplasia (ADM)
Cyst
Pancreatic intraepithelial neoplasia (PanIN)
Intraductal papillary mucinous neoplasm (IPMN)
Surrogate markers of pancreatic cancer
Focal fatty replacement
Lobulocentric atrophy
Duct dilatation/Cyst/IPMN

**Table 2 cancers-13-00686-t002:** Cancer-related pathological changes in the pancreas of various animals.

Hamster	Precursor Lesions	
BOP	PanIN	Takahashi, 2018
BOP, ethionine-methionine	PanIN	Mizumoto, 1989
Bile reflux into the pancreatic duct	IPMN	Adachi, 2006
**Mouse**		
LSL-*Kras*^G12D^; *Pdx1*-Cre	PanIN	Hingorani, 2003
LSL-*Kras*^G12D^; *Ptf*/p48-Cre	PanIN	Hingorani, 2003
KIC model, *Pdx*-Cre; LSL-*Kras*^G12D^; Ink4a/Arf^lox/lox^	PanIN	Aguirre, 2003
Pdx-Cre; LSL-*Kras*^G12D^; *Trp53*^lox/lox^; Ink4a/Arf^lox/lox^	PanIN	Bardeesy, 2006
KPC model, *Pdx1*-Cre; LSL-*Kras*^G12D^; LSL-*Trp53*^R172H/+^	PanIN	Hingorani, 2005
KD model, *Pdx*-Cre; LSL-*Kras*^G12D^; *SMAD*4^lox/lox^	PanIN, IPMN	Kojima, 2007
*Ptf1a*-Cre; LSL-*Kras*^G12D^; *SMAD4*^lox/lox^	PanIN, MCN	Izeradjene, 2007
*Ptf1a*-Cre; LSL-*Kras*^G12D^; *Notch1*^lox/lox^	PanIN	Hanlon, 2010
*Ptf1a*-Cre; LSL-*Kras*^G12D^; *Tgfr2*^lox/lox^	PanIN	Ijichi, 2006
*Pdx1*-Cre; LSL-*Kras*^G12D^; *Brca2*^tr/Δ11^	PanIN	Skoulidis, 2010

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
