# Peer review of "Pathological Changes in Pancreatic Carcinogenesis: A Review"

_cancers, 2021, doi:10.3390/cancers13040686_

Round 1

Reviewer 1 Report

This manuscript provides a brief overview on pathological changes in pancreatic carcinogenesis.

An abstract is missing (the abstract section contains only general guidelines for writing an abstract).

Introduction, first paragraph: "Pancreatic ductal adenocarcinoma is the leading cause of cancer-related deaths worldwide"?

Introduction, second paragraph: "centro-acinar resion"? "condired to arise"?

Pathology of Pancreatic Tumors, Primary Pancreatic Tumors: "Primary pancreatic tumors are observed in 3% of cases, and most primary tumors are PDACs (2%), followed by neuroendocrine tumors (0.3%) and intraductal papillary mucinous neoplasms (0.1%). High-grade PanINs are detected in 0.04% of the cases. Occult PDAC is discovered incidentally in 8% of PDAC cases. Distant metastasis is an important prognostic factor for PDAC. Twenty-seven percent of occult cancer cases involve distant metastasis, and most of them were localized in the pancreatic tail. Asymptomatic cancers of the pancreatic tail have a greater ability to metastasize to distant sites than those of the pancreatic head and body." – Citations are needed for these numbers.

Pathology of Pancreatic Tumors, Metastatic Pancreatic Tumors: "In the same cohort, secondary pancreatic tumors resulting from invasion and metastasis of malignant tumors were detected in 175 cases (2%). The primary sites of metastatic tumors were the stomach (17%), lung (17%), colorectum (6%), esophagus (5%), gall bladder (3%), and kidney (2%). The primary sites of direct invasion were the stomach (19%), common bile duct (3%), gall bladder (1%), and liver (1%). The infiltration rate of hematologic tumors was 16%." – Which "cohort" is meant here?

Figure 1: Power of magnification and staining technique should be given for the histological images.

Pancreatic Intraepithelial Neoplasia (PanIN), second paragraph: "A lot pf 99 studies"?

Pancreatic cystic lesions, last paragraph: "However, high-grade PanINs underwent autopsy are often detected in the branch pancreatic duct"?

Line 175: "Tgfr2" -> Tgfbr2? "allelle" -> allele.

Author Response

We deeply express our gratitude to the Editor and referees for devoting their time and making significant efforts to carefully review our manuscript. We strongly believe that our paper has considerably benefitted from your valuable and insightful comments. This response letter contains point-by-point responses to each of the comments made by the reviewer.

  1. This manuscript provides a brief overview on pathological changes in pancreatic carcinogenesis.

An abstract is missing (the abstract section contains only general guidelines for writing an abstract).

Introduction, first paragraph: "Pancreatic ductal adenocarcinoma is the leading cause of cancer-related deaths worldwide"?

Introduction, second paragraph: "centro-acinar resion"? "condired to arise"?

Response: Thank you for pointing out this issue. We have added an abstract at the respective location in the revised manuscript. We have also corrected the incorrect spelling of the above-mentioned words.

  1. Pathology of Pancreatic Tumors, Primary Pancreatic Tumors: "Primary pancreatic tumors are observed in 3% of cases, and most primary tumors are PDACs (2%), followed by neuroendocrine tumors (0.3%) and intraductal papillary mucinous neoplasms (0.1%). High-grade PanINs are detected in 0.04% of the cases. Occult PDAC is discovered incidentally in 8% of PDAC cases. Distant metastasis is an important prognostic factor for PDAC. Twenty-seven percent of occult cancer cases involve distant metastasis, and most of them were localized in the pancreatic tail. Asymptomatic cancers of the pancreatic tail have a greater ability to metastasize to distant sites than those of the pancreatic head and body." – Citations are needed for these numbers.

Response: Thank you for the suggestion. We have added the respective references for these data in the revised manuscript.

  1. Pathology of Pancreatic Tumors, Metastatic Pancreatic Tumors: "In the same cohort, secondary pancreatic tumors resulting from invasion and metastasis of malignant tumors were detected in 175 cases (2%). The primary sites of metastatic tumors were the stomach (17%), lung (17%), colorectum (6%), esophagus (5%), gall bladder (3%), and kidney (2%). The primary sites of direct invasion were the stomach (19%), common bile duct (3%), gall bladder (1%), and liver (1%). The infiltration rate of hematologic tumors was 16%." – Which "cohort" is meant here?

Response: Thank you for the valuable comment. We have deleted the phrase "In the same cohort” in the revised manuscript.

  1. Figure 1: Power of magnification and staining technique should be given for the histological images.

Response: Thank you for the advice. We have added details of the power of magnification and staining technique in the legend of Figure 1.

  1. Pancreatic Intraepithelial Neoplasia (PanIN), second paragraph: "A lot pf 99 studies"?

Response: Thank you for pointing out the issue. We have corrected the incorrect spelling.

  1. Pancreatic cystic lesions, last paragraph: "However, high-grade PanINs underwent autopsy are often detected in the branch pancreatic duct"?

Response: In our autopsy study, high-grade PanINs were located in the branch pancreatic duct. We have added the respective reference.

  1. Line 175: "Tgfr2" -> Tgfbr2? "allelle" -> allele.

Response: We apologize for the incorrect spelling. We have corrected the spelling in the revised manuscript.

Reviewer 2 Report

The manuscript entitled “Pathological Changes in Pancreatic Carcinogenesis: A Review” is an interesting literature revision on the role of pathological changes in the pancreas of humans and experimental animals during carcinogenesis. The manuscript is well written and balanced; anyway, the following revisions are warranted.

- English should be properly revised

- Please correct the mistyping in the text

- Abstract should be added

- References 1-2 should be updated

- Paragraph 2.2: Authors should deepen several aspects of metastatic findings; in particular, since pancreatic cancer is a stromal cancer, Authors should define the role of stromal component vs angiogenetic implications (Authors can refer to Angiogenesis in pancreatic ductal adenocarcinoma: A controversial issue. Oncotarget. 2016 Sep 6;7(36):58649-58658. doi: 10.18632/oncotarget.10765)

- Paragraph entitled “Pancreatic Cystic Lesions”: Authors should better explain that from these precursor lesions do not arise the classical pancreatic cancer phenotype (pancreatic ductal adenocarcinoma), but rarer cystic pancreatic cancers

Author Response

We deeply express our gratitude to the Editor and referees for devoting their time and making significant efforts to carefully review our manuscript. We strongly believe that our paper has considerably benefitted from your valuable and insightful comments. This response letter contains point-by-point responses to each of the comments made by the reviewer.

  1. The manuscript entitled “Pathological Changes in Pancreatic Carcinogenesis: A Review” is an interesting literature revision on the role of pathological changes in the pancreas of humans and experimental animals during carcinogenesis. The manuscript is well written and balanced; anyway, the following revisions are warranted.

- English should be properly revised

- Please correct the mistyping in the text

- Abstract should be added

Response: Thank you for pointing this out. We have corrected the spelling. We have also included an abstract.

  1. - References 1-2 should be updated

Response: We have updated the references in the revised manuscript.

  1. - Paragraph 2.2: Authors should deepen several aspects of metastatic findings; in particular, since pancreatic cancer is a stromal cancer, Authors should define the role of stromal component vs angiogenetic implications (Authors can refer to Angiogenesis in pancreatic ductal adenocarcinoma: A controversial issue. Oncotarget. 2016 Sep 6;7(36):58649-58658. doi: 10.18632/oncotarget.10765)

Response: According to your suggestion, we have detailed several aspects of metastatic tumors.

  1. - Paragraph entitled “Pancreatic Cystic Lesions”: Authors should better explain that from these precursor lesions do not arise the classical pancreatic cancer phenotype (pancreatic ductal adenocarcinoma), but rarer cystic pancreatic cancers

Response: Thank you for your suggestion. We have explained pancreatic cystic lesions and cystic pancreatic cancers.